# Experimental Study on Timber−Lightweight Concrete Composite Beams with Ductile Bolt Connectors

**DOI:** 10.3390/ma14102632

**Published:** 2021-05-18

**Authors:** Yafeng Hu, Yang Wei, Si Chen, Yadong Yan, Weiyao Zhang

**Affiliations:** College of Civil Engineering, Nanjing Forestry University, Nanjing 210037, China; hill_hu_2003@163.com (Y.H.); cs0714@njfu.edu.cn (S.C.); sdzcyyd@163.com (Y.Y.); zwy1230419@163.com (W.Z.)

**Keywords:** timber, composite beam, lightweight concrete, connectors, bending performance

## Abstract

A timber–lightweight−concrete (TLC) composite beam connected with a ductile connector in which the ductile connector is made of a stainless−steel bolt anchored with nuts at both ends was proposed. The push−out results and bending performance of the TLC composite specimens were investigated by experimental testing. The push−out results of the shear specimens show that shear–slip curves exhibit good ductility and that their failure can be attributed to bolt buckling accompanied by lightweight concrete cracking. Through the bending tests of ten TLC composite beams and two contrast (pure timber) beams, the effects of different bolt diameters on the strengthening effect of the TLC composite beams were studied. The results show that the TLC composite beams and contrast timber beams break on the timber fiber at the lowest edge of the TLC composite beam, and the failure mode is attributed to bending failure, whereas the bolt connectors and lightweight concrete have no obvious breakage; moreover, the ductile bolt connectors show a good connection performance until the TLC composite beams fail. The ultimate bearing capacities of the TLC composite beams increase 2.03–3.5 times compared to those of the contrast beams, while the mid-span maximum deformation decrease nearly doubled.

## 1. Introduction

Currently, increasing attention is being given to the reinforcement of timber structures, and many scholars have proposed reinforcement methods for timber structures [1,2]. The TLC composite beam is a kind of structural component formed by combining lightweight concrete and timber, and it is not only used for new structures but also for reinforcement of existing timber structures [3,4,5,6,7,8,9,10,11,12,13,14,15,16,17]. The TLC composite beam consists of a lightweight concrete flange supported by a timber web, which is attached by different types of connectors. The TLC composite beam makes full use of the characteristics of different materials, the timber beam, and the lightweight concrete slab bear compression and tension, respectively. Under ideal conditions, the timber primarily sustains tensile stress, and the lightweight concrete bears compression stress generated by the moments and composite action. Compared to timber structures, the lightweight concrete slab can increase the cross−sectional stiffness and reduce the weight of the TLC composite beam. Moreover, the TLC composite beam maintains the aesthetic appearance of timber structures. The materials used in TLC composite beams are environmentally friendly and resource saving.

The effect of the connection determines the working performance of the composite structure [18,19,20,21,22]. The performance of timber–concrete composite beams with different types of connections has been reported by existing studies [14,15,16,23,24,25]. The connection stiffness affects the degree of composite action between the upper and lower parts of composite beams, and the bending behavior of the TLC composite beam is mainly determined by the connectors: (i) the shear stiffness of the connector affects the stiffness of the TLC composite beam; (ii) the shear bearing capacity of the connector determines the shear bearing capacity of the interface and the bending capacity of the TLC composite beam; and (iii) the ductility of the TLC composite beam is affected by the deformation capacity of the connectors before ultimate failure.

In addition to considering the bearing capacity, the ductility of TLC composite beams is often the focus of engineering design and application. When the timber fiber reaches its maximum tensile stress or strain, the TLC composite beam will be brittle even if the connector still shows elasticity. In contrast, when the yield of the connector occurs before timber collapse, the TLC composite beam produces plastic deformation and shows nonlinear and ductile behavior [17,26]. Compared with ordinary timber–concrete composite beams, TLC composite beams exhibit a lower bearing capacity and better ductility due to the properties of lightweight concrete. However, less research has been carried out on TLC composite beams.

In this paper, a novel ductile bolt connector was proposed for a TLC composite beam, in which the bolt connectors are expected to provide ideal ductility by bending deformation and ensure reliable anchoring through end nut anchoring, and the proposed connector is simple, easy to obtain, and economical. To investigate the mechanical performance of TLC composite beams with ductile bolt connectors, push−out tests of shear specimens and four−point bending tests of TLC composite beams were carried out.

## 2. Materials and Experiments

### 2.1. Material Properties

The shear specimens and TLC composite beams were made of the same materials, including Scotch Pine profile bars (Table 1), lightweight concrete, and ductile bolt connectors. Based on the test results of Scotch Pine wood, the ultimate tensile strength (f_bt_), ultimate compression strength (f_bc_), and elastic modulus were 81.0 MPa, 33.4 Mpa, and 10.43 GPa, respectively. The light aggregate was shale ceramicite produced in Yichang Everbright Ceramic Products Co., Ltd, Yi Chang, China, and its aggregate sizes were constituted of 0–3 mm, 3–5 mm, and 5–20 mm at a ratio of 2:2:6. The mixture ratio of lightweight concrete was 1:0.2:0.65:1.65:0.015:0.01 (cement:fly ash:water:lightweight aggregate:water reducer:glass fiber) (Table 2). The compression strength of lightweight concrete cylinders (ϕ150 × 300 mm) was 22.2 MPa, and the elastic modulus at the testing time was 18.4 GPa. The yield strength and elastic modulus of the bolt connectors with different diameters were between 514.0–520.0 MPa and 111.2–123.7 GPa.

### 2.2. Shear Specimens

For the mechanical analysis of the TLC composite beam, the mechanical properties of the shear specimen must be determined in advance. Referring to EN 1995−1 [26], three shear specimens with the same parameters were designed for different bolt diameters. The specimens consist of one timber slab (140 mm × 360 mm × 70 mm) and two lightweight concrete slabs (140 mm × 360 mm × 70 mm). The timber slab was located in the middle of the specimen, and the lightweight concrete slabs were symmetrically arranged on both sides of the timber slab [26]. The timber slab and the lightweight concrete slabs were connected with ductile bolts. The diameters of the bolts in the shear specimen were 6 mm, 8 mm, 10 mm, 12 mm, and 16 mm, and the specimen groups were named SC6, SC8, SC10, SC12, and SC16 according to the diameters of the bolts. The bolts were embedded 50 mm in lightweight concrete. The details are shown in Figure 1.

An electric−hydraulic serving machine with a maximum loading force of 2000 kN and a TDS collection system were used in the push−out test. To eliminate the influence of gaps between timber and lightweight concrete and confirm that the whole device works properly, before the formal test, a 2 kN load was applied for preloading. Displacement control was adopted, and the loading rate was set as 1.0 mm/min. The relative slip was measured by a linear variable displacement transducer (LVDT) (Beijing King Sensor Technology Co., Ltd, Beijing, China). Six LVDTs were arranged at the bottom, middle, and top of the shear specimen, as shown in Figure 1b,c.

### 2.3. Composite Beams

Two beam specimens with the same parameters were manufactured for one group, as shown in Table 3. Apure timber beam used as the benchmark beam, which is named group B0. The TLC composite beams were divided into five groups according to the diameter of the bolt with 6 mm, 8 mm, 10 mm, 12 mm, and 16 mm, and the corresponding names were called group BP6, group BP8, group BP10, group BP12, and group BP16. The central spacing of bolts is 150 mm. The section shape of the TLC composite beam is T−shaped, with a timber slab in the bottom part (web) and a lightweight concrete slab on the top part (flange) [15,27]. The size of the web is 70 mm × 140 mm × 3000 mm, and the size of the flange is 200 mm × 70 mm × 3000 mm. In the manufacturing process of the TLC composite beam (Figure 2), first, the anchor holes were drilled through the timber beam; then, the forms were built according to the size of the flange, and the structural reinforcements were assembled; in the third, the ductile bolts were embedded into the holes of the timber web; finally, the mixed concrete was poured to form a flange, and the TLC composite beams were completed after removing the form.

A four−point bending test was used to investigate the bending performance of the TLC composite beam. The distance between the neutral lines of the two fixed supports is 2700 mm, and the distance from one loading point to the same side support is 900 mm. The bending load of the TLC composite beam is controlled by displacement, and the loading rate is 2 mm/min. A laser displacement meter (LDM) and two LVDTs were set at the mid-span and the supports to measure the vertical displacement of the TLC composite beam [28]. To measure the relative slip of the interface between timber and lightweight concrete, four LVDTs (D1, D2, D3, and D4) were arranged, and eight strain gauges were pasted on the mid-span cross−section to measure the strain of lightweight concrete and timber. The specific layout is shown in Figure 3.

## 3. Results and Discussion

### 3.1. Shear Specimens

#### 3.1.1. Failure Modes

The typical failure modes of the shear specimens are shown in Figure 4. When the push−out load value is less than 40% of the ultimate load, there is no obvious destruction phenomenon, where only slight cracks are gradually formed at the interface between timber and lightweight concrete, the slip of the interface between timber and concrete is extremely small, and the stiffness of the shear specimen is relatively high. When the load reaches approximately 40–70% of the ultimate load, the inclined shear cracks extend downward from the middle part of the lightweight concrete, and then the lightweight concrete peels off. When the load reaches approximately 70–100% of the ultimate load, the relative slip increases sharply. There was no significant vertical separation between the timber and the lightweight concrete during the push−out test. After the test, lightweight concrete was crushed at the position of the bolt. The bolt bending deformation occurred significantly near the interface of the timber, and the size of the anchor hole increased along the loading direction. The bond between bolt and timber remained intact. In general, the shear specimens can suffer a large plastic deformation during the entire loading process and withstand a high load level, and the failure modes of the shear specimens connected with ductile bolts may be attributed to ductile failure.

#### 3.1.2. Load–Slip Curves

The load–slip curves of the shear specimens are shown in Figure 5. The load–slip curve can be divided into three stages. At the first stage (0–15% ultimate load), the curves are generally linear with increasing load. In this process, the bolt and the timber are well bonded, the lightweight concrete provides effective shear transfer for the bolts, and the slip curve develops slowly and steadily. In the second stage (15–100% ultimate load), the increase rate of the interface slip is obviously accelerating, and the load–slip relationship is nonlinear because the lightweight concrete is gradually cracked and the bending deformation of the bolt obviously occurs as the shear load increases. At the third stage (after ultimate loading), this process is mainly shown as the load decrease and the slip increased sharply until the shear specimen fails. It can be seen from the load–slip curve (Figure 5) that when the bolt diameter increases from 6 mm to 16 mm, the bearing load of the shear specimens increases from 27.54 kN (SC6) to 60.23 kN (SC16), and the corresponding slip at the maximum load shows a trend of gradual decrease from 15.0 mm to 6.2 mm. The bearing capacity increases significantly, and the slip decreases as the diameter of the bolt increases for the shear specimens, indicating that the bond properties between timber and concrete are good. It should be noted that the slip before the peak is highly significant, and the load–slip response of the ductile connector shows a good bearing capacity and ductility.

#### 3.1.3. Shear Stiffness and Ductility Coefficient

The shear stiffness *K* (Formula (1)) is an important index for evaluating the connecting performance of shear specimens. To reflect the stiffness of shear specimens at different loading stages, the stiffness corresponding to 40%, 60%, and 80% of the ultimate load (*P*_u_) is defined as *K_S_*_,0.4_, *K_S_*_,0.6_, and *K_S_*_,0.8_ to reflect the serviceability limit state, ultimate limit state, and critical failure. The ductility coefficient *D* (Formula (2)) is defined to quantitatively describe the working ductility of shear connectors.
(1)K=P/S
(2)$$D=S_{u}/S_{y}$$
where *K* is the shear stiffness of the specimen, kN/mm; *P* is the load, kN; *S* is the slip, mm; *S_u_* is the slip when the corresponding load drops to 80% of the ultimate load *P*_u_, mm; and *S_y_* is the slip when the corresponding load rises to 80% of the ultimate load *P*_u_, mm.

The test results of the TLC shear specimens with ductile bolts are shown in Table 4. We can see that with the increase of the bolt diameter, the shear stiffness increased significantly, among which, *K_S_*_,0.4_, *K_S_*_,0.6_, and *K_S_*_,0.8_ increased from 3.92 kN/mm, 2.49 kN/mm, 2.07 kN/mm of shear specimens with 8 mm diameter bolt to 22.52 kN/mm, 18.69 kN/mm, 14.11 kN/mm of that with 16 mm diameter bolt, when the shear specimens with a maximum diameter of bolts exhibit a maximum shear stiffness. The average ductility coefficient (*D*) is between 1.71 (SC6) and 2.89 (SC8), and the ductility coefficient of the other shear specimens is between 2.29 and 2.89 with little change. The average ultimate displacement when the load drops to 80% of the ultimate load is between 7.85 mm and 21.05 mm; in general, the bolt diameter increases, and its value decreases. The significant ultimate displacement indicates a good deformation capacity when the bolt diameter is relatively large. Although the ductility coefficient of the shear specimen is not high, it is related to the rapid decline of the bearing capacity of the shear specimen after its load reaches its peak value.

### 3.2. Composite Beams

#### 3.2.1. Observations and Failure Modes

The B0 group is the control beam made of pure timber, and its size is the same as the timber part in the TLC composite beam. For the control beam, failure occurred from rupturing at the bottom in the mid-span region, and the timber ruptured in the tension zone (Figure 6). The bending deformation is obvious at the failure. The typical failure modes of the TLC composite beams are shown in Figure 7. For the TLC composite beams, the lightweight concrete flange and the timber web can work together before finally failing. The failure of the TLC composite beam is manifested as timber fracture in the tensile zone at the bottom of the web, and the lightweight concrete part in the whole loading process is basically intact. The global bending deformation of the TLC composite beam corresponding to the final failure is significant, showing good deformation capacity and satisfactory ductility of the structure. Taking BP6 group test beams as an example, the mid-span deflection corresponding to the maximum load is 69 mm (approximately *L*/45, *L* is the span).

#### 3.2.2. Flexural Stiffness and Capacity

The bending test results of the control beams (B0) and the TLC composite beams (BP6–BP16) are shown in Table 5. The mid-span displacement of the test beams at a 10 kN load is listed in the table. The average mid-span displacement of the control beams is 41.8 mm and that of the TLC composite beams is between 7.8 mm and 16.6 mm. Compared with the control beam, the rigidity of the TLC composite beam increases 2.51–7.38 times. According to Euro code 5 [19], when a timber beam is used as a floor beam or slab, the displacement limit corresponding to the serviceability limit state is specified as *L*/300 (*L* is the span). The results showed that under the *L*/300 mid span displacement, the load *P**_L_*_/300_ corresponding to the service limit state of the composite beam is 2.61–6.12 times higher than that of the timber beam. In addition, the *P_L/_*_300_/*P*_max_ values of the timber beam and composite beam were 0.18 and 0.21–0.34, respectively, which showed that the composite beam had a high efficiency of bearing capacity utilization [29]. The combination efficiency coefficient *E* can be used to describe the combination degree of the combination structure. *E* = 1 indicates a complete combination, and *E*=0 indicates a noncombination structure. A larger value of *E* (Formula (3)) corresponds to a higher combination degree of the structure with a greater stiffness and bearing capacity.

According to the bending test results of the control beams and the TLC composite beams, the load–displacement test curves of all beams can be obtained, as shown in Figure 8. The bearing capacity of all TLC composite beams is doubled compared with that of the control beam, while the mid-span maximum displacement of the TLC composite beam decreases gradually with increasing bolt diameter. The gradient of the load–displacement curve in the figure of the test results reflects the stiffness of the TLC composite beam. By comparing the analytical stiffness with the test curve, it can be seen that there is good correlation between the two curves.
(3)$$E=\frac{\Delta_{NC}-\Delta_{PC}}{\Delta_{NC}-\Delta_{FC}}$$
where Δ is mid-span displacement; subscripts *NC*, *FC*, and *PC* represent no combination, complete combination, and partial combination, respectively.

The combined efficiency coefficients at different load levels are given in Table 5. When *P* = 10 kN, *E* = 0.53–0.76; when Δ = *L*/300, *E* = 0.62–0.72. With a larger bolt diameter, a larger *E* value can be obtained. The results showed that the mechanical properties of the composite beams were between those of the complete composite and partial composite. With a larger bolt diameter, a greater rigidity of the TLC composite beam and a higher composite efficiency can be achieved.

By comparing with the control beams (B0), the “*E*” value, the improvement of its bearing capacity, the mid-span displacement and other properties of the TLC composite beams (BP6–BP16) at each stage can be obtained. Specifically, (i) in the three stages of “*P* = 10 kN”, “Δ = *L*/300” and “Timber cracking state”, the “*E*” value of each TLC composite beam decreases with increasing load value. In addition, in the same stage, as the bolt diameter of the TLC composite beam (except for the TLC composite beam BP16) increases from 6 mm to 16 mm, the “*E*” value increases first and then decreases. (ii) The ultimate bearing capacity of the TLC composite beams (BP6–BP16) is increased by 2.03–3.50 times compared with control beams (B0) in the final state. The ultimate bearing capacity of the TLC composite beams has no obvious correlation with the bolt diameter. Therefore, the ultimate bearing capacity of each TLC composite beam is mainly determined by the performance of timber. (iii) The variation of the mid-span displacement of a TLC composite beam, in the “*P* = 10 kN” stage, its value is much lower than the control beam (B0), and the displacements of composite beams decrease with the bolt diameter increased significantly (dropped from 16.6 mm to 7.2 mm), in the normal stage, the displacement under the control of the bolt diameter, namely, the bolt diameter increases, the displacement decreased. In the “Timber cracking state” stage, when the bolt diameter is smaller (≤8 mm), the displacement of the TLC composite beam (BP6 and BP8) and control beam (B0) are closer, the lightweight concrete is intact when the TLC composite beams fractured, like the control beam, due to the timber fiber fracture times; when the bolt diameter is larger (≥10 mm), the stiffness of the TLC composite beams (BP16–BP16) compared with the control beam (B0) increase about beam to 7 times the bolt diameter is governed by the bolt displacement of the TLC composite beam.

#### 3.2.3. Analysis of Section Strain and Interface Slips

The strain change trend on the middle span section of the TLC composite beam (Figure 9) was approximately similar [30]. When the load is less than 15% of the maximum load, the strains are linear. As the load increases (15% to 100% maximum load period), the strain gradually exhibits a nonlinear trend; therefore, the interface slips of the TLC composite beam are nonlinear. Meanwhile, as the bolt diameter increases, the strains of the BP6 group to the BP16 group gradually decrease. Among them, the compressive strain of lightweight concrete for the flange plate decreases from 4000 με to 2200 με, and the tensile strain of timber for the web plate decreases from 4900 με to 1180 με. When the bolt diameter is the smallest (BP6), in the lightweight concrete section of the TLC composite beam, the strain is in the state of full compression [30]. Under the same load, the compressive strain of the TLC composite beam flange decreases gradually with increasing bolt diameter, and the maximum compressive strain range is from 3750 με to 2200 με. At the whole stage of loading, part of the flange (lightweight concrete) is in a complete compression state; part of the web (timber), with increasing bolt diameter, the neutral axis of the TLC composite beam moves toward the lower edge of the web, and the strains of the upper edge of the web change to compression [31]. In this state, the maximum tensile strain of timber in the web of the TLC composite beam is close to the maximum tensile strain of timber measured by the tension test, which indicates that the strength of timber has been fully utilized. The maximum compressive strain value of the lightweight concrete of the TLC composite beam is generally less than 2000 με, while the main failure of the TLC composite beam is the fracture of timber in the web. Therefore, it is difficult to give full play to the strength of lightweight concrete. During the whole loading process, the strain curve of the TLC composite beam section is smooth, and the strain distribution has no distortion, which reflects the good deformation coordination between timber and lightweight concrete.

Figure 10 shows the load–slip curves measured at the interface between the mid–span and the support of the TLC composite beams. In the early stage of loading (<5 kN), the interface slip of the TLC composite beams is very small (<0.5 mm). When the load reaches approximately 10 kN (approximately 40% of the ultimate load), the slip increases gradually with increasing load; Table 6 shows the relative slip value of the TLC composite beam interface under different load levels. The relative slip value corresponding to the mid-span displacement of *L*/300 ranges from 0.31 mm to 1.19 mm; the relative slip value corresponding to the mid-span displacement of *L*/250 ranges from 0.39 mm to 1.20 mm. The results show that the relative slip increases gradually from the mid-span to the support; the load–slip curve changes linearly in the early stage of loading (<5 kN), and the relative slip value decreases significantly with increasing bolt diameter. In the later stage of loading, with the slow increase in load, the load–slip curve develops stably.

### 3.3. Theoretical Analysis

According to the test results and referring to Euro code 5, for the structure composed of timber and lightweight concrete through bolt connectors (see Figure 11), the section stiffness can be calculated by the equivalent section stiffness method (*EI_y_*)*_ef_* (Formula (4)). The connection coefficient *γ_b_* (Formulas (5) and (6)) is used to describe the stiffness reduction caused by the interface slip between timber and lightweight concrete. The range of *γ_b_* is between 0 and 1. *γ_b_* = 0 indicates that the material interface is not connected, and *γ_b_* = 1.0 indicates that the material interface is rigid. With the larger value of *γ_b_*, the larger the anti-sliding stiffness of the joint can be obtained. The stiffness formula is as follows [32].
(4)$${\left(EI_{y}\right)}_{e}{}_{f}=E_{1}(\frac{h_{1}{}^{3}\cdot b_{1}}{12}+\gamma_{b}\cdot A_{1}\cdot a_{1}{}^{2})+E_{2}(\frac{h_{2}{}^{3}\cdot b_{2}}{12}+A_{2}\cdot a_{2}{}^{2})$$

In Formulas (5) and (6): *E*_1_ is the elastic modulus of lightweight concrete, N/mm^2^; *E*_2_ is the elastic modulus of timber, N/mm^2^; *n_c_* is the section conversion coefficient between lightweight concrete and timber, *n_c_* = *E*_1_/*E*_2_; *A*_1_ is the section area of lightweight concrete, mm^2^; *A*_2_ is the section area of timber, mm^2^; *h*_1_ is the height of the lightweight concrete flange, mm; *h*_2_ is the height of the timber web, mm; *a*_1_ is the distance from the center of the lightweight concrete to the neutral axis, mm; *a*_2_ is the distance from the center of the timber web to the neutral axis, mm; *b*_1_ is the width of the lightweight concrete flange; and *b*_2_ is the width of the timber web, mm.
(5)$$\gamma_{b}=\frac{1}{1+k}$$
(6)$$k=\frac{\pi^{2}\cdot E_{1}\cdot A_{1}\cdot s_{i}}{K\cdot l^{2}}$$
where *s_i_* is the effective spacing of shear connectors, mm; *K* is the shear slip stiffness of shear connectors, N/mm; *l* is the span of the beam, mm; and *k* is a dimensionless quantity that reflects the combined influence of the slip stiffness and spacing of connectors, lightweight concrete cross−sectional area and material properties on the connection coefficient.

In Formula (4), *a*_1_ and *a*_2_ are calculated by Formulas (7) and (8), respectively.
(7)$$a_{2}=\frac{n_{c}\gamma_{b}A_{1}(h_{1}+h_{2})}{2(n_{c}\gamma_{b}A_{1}+A_{2})}$$
(8)$$a_{1}=\frac{h_{1}+h_{2}}{2}-a_{2}$$

Ignoring the influence of shear force and axial force, the formula for calculating mid-span displacement of bending specimens under four−point loading can be obtained in this paper.
(9)$$\Delta =\frac{Pa}{48{\left(EI_{y}\right)}_{e}{}_{f}}(3l^{2}-4a^{2})$$
where Δ (Formula (9)) is the mid-span displacement of the specimen, mm; *P* is the total load of two loading points, N; (*EI_y_*)*_ef_* is the equivalent section stiffness, N·mm^2^; *l* is the span of the beam, mm; and *a* is the distance between the loading point, mm.

According to Formulas (4)–(9), we can calculate the cross−section parameters and stiffness of the TLC composite beam, and the calculation results can be analyzed with Figure 11 and Table 7 [27]. Substituting the results of the numerical analysis in Table 7 into Formulas (5) and (6), *γ_b_* is obtained. The equivalent section stiffness (*EI_y_*)*_ef_* of the TLC composite beams is calculated by substituting *γ_b_* into Formula (5), and the results are shown in Table 7. The connection coefficient *γ_b_* is between 0.07–0.37, and the stiffness is between 2.09 × 10^11^ kN·mm^2^–5.96 × 10^11^ kN·mm^2^; therefore, the connection coefficient and stiffness increase with bolt diameter, and the connection coefficient and stiffness are increased by 5 times and 3 times, respectively. Here, *a*_1_ and *a*_2_ are the distances from the centerline of lightweight concrete and timber to the cross−section neutral axis, and their sum is a constant value. According to Formulas (7) and (8), the numerical results of *a*_1_ and *a*_2_ are shown in Table 7. It can be seen that the neutral axis of the TLC composite beam cross−section is always on the timber (web), and its position is approximately 35–65 mm at the lower edge of lightweight concrete (flange plate). With the increase in bolt diameter, the neutral axis gradually approaches the lightweight concrete with the increase in bolt diameter.

## 4. Conclusions

In this paper, the shear and bending resistances of timber–lightweight−concrete specimens with bolts of different diameters were studied. Five groups of push−out specimens and six groups of beam specimens (including one group of contrast beam specimens) were tested for shear and bending resistance, respectively. Through experimental study and analysis, the following conclusions can be drawn:(1)It was observed that the typical failure mode of shear specimens belongs to ductile failure, which was shown as bolt buckling accompanied by lightweight concrete cracking;(2)From the push−out test, the load–slip curve of shear specimens can be divided into three stages: (i) elastic stage, the elastic modulus and stiffness increased with the bolt diameter; (ii) elastic−plastic stage, in this stage, the load increased slowly, and the stiffness decreased gradually. When the bolt diameters were 8–12 mm, the specimens showed greater bearing capacity and plastic deformation; and (iii) descending stage, mainly due to the instability of lightweight concrete. Generally, the bearing capacity increased with increasing bolt diameter (27.54–60.23 kN), but the slip value decreased with increasing bolt diameter (15.0 mm to 6.2 mm);(3)When the bolt diameter was 6–16 mm, the displacement and stiffness of the TLC composite beams were evenly distributed, the connection coefficient (*γ_b_*) was dispersed in the range of 0.07 to 0.37, and the stiffness of the TLC composite beams was between 2.09 × 10^11^ kN·mm^2^–5.96 × 10^11^ kN·mm^2^;(4)Shear cracks were found in the web (timber beam) and the flange (lightweight aggregate concrete panel), and their morphology had no significant change before and after the failure of the TLC composite beam. The fiber rupture in the middle of the web (timber beam) was the main reason for the bending failure of the TLC composite beam. The average bearing capacity of the control group was 12.8 kN, that of the TLC composite beams was 27.25–39.1 kN, and the bearing capacity of the timber beam was significantly improved;(5)The bending test showed that the load–deflection curves of bolted TLC composite beams were basically consistent with the calculated formula, and the strain distributions on the section height of the TLC composite beams were uniform. This TLC composite structure is an ideal technology to strengthen pure timber beams.

## Figures and Tables

**Figure 1 materials-14-02632-f001:**
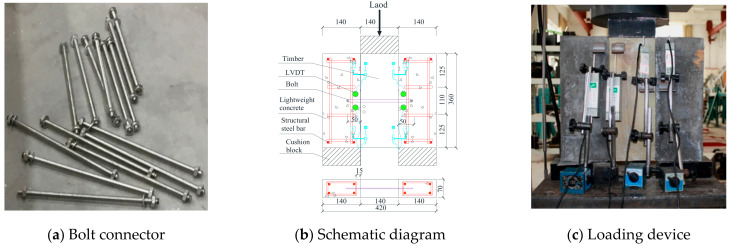
Bolt and push−out test schematic.

**Figure 2 materials-14-02632-f002:**
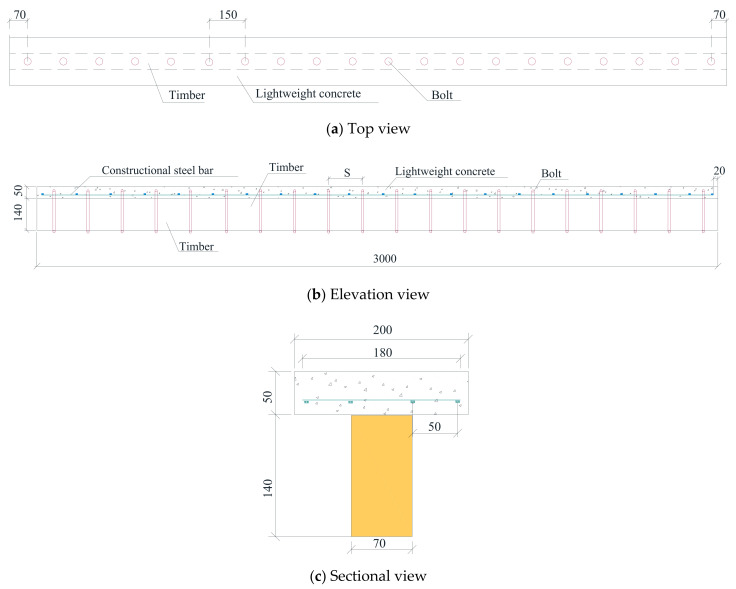
Structure schematic of the TLC composite beams.

**Figure 3 materials-14-02632-f003:**
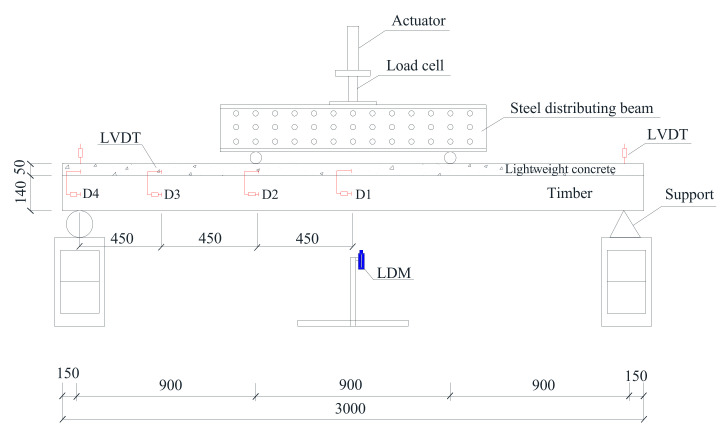
Four−point bending test of the TLC composite beam.

**Figure 4 materials-14-02632-f004:**
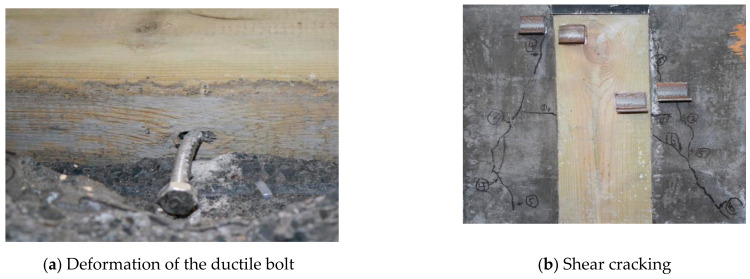
Typical failure modes of the shear specimens.

**Figure 5 materials-14-02632-f005:**
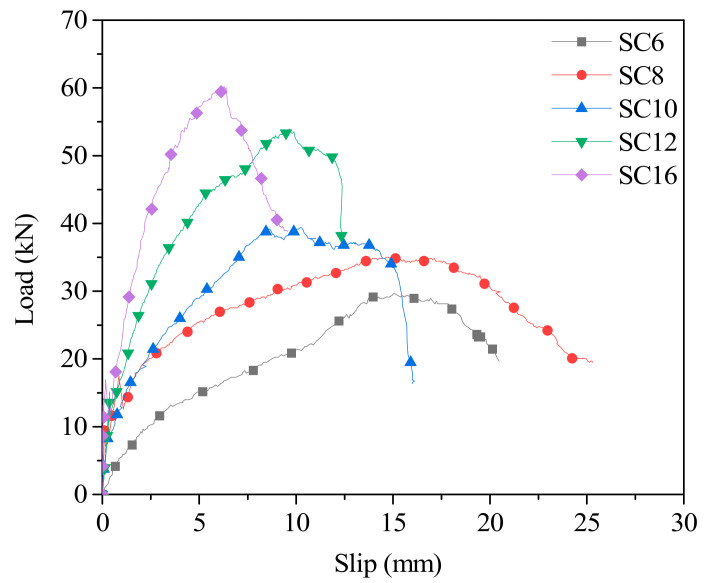
Representative load–slip curves of the push–out test for the TLC shear connections with ductile bolts.

**Figure 6 materials-14-02632-f006:**
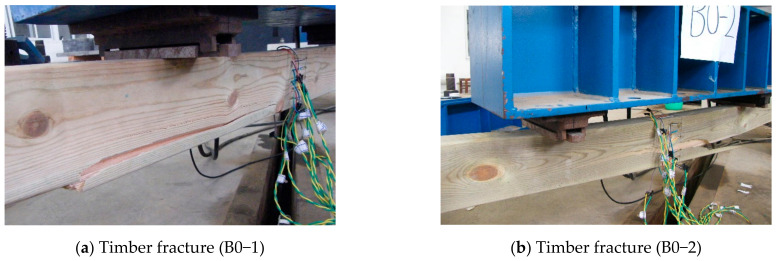
Typical failure modes of the control beams.

**Figure 7 materials-14-02632-f007:**
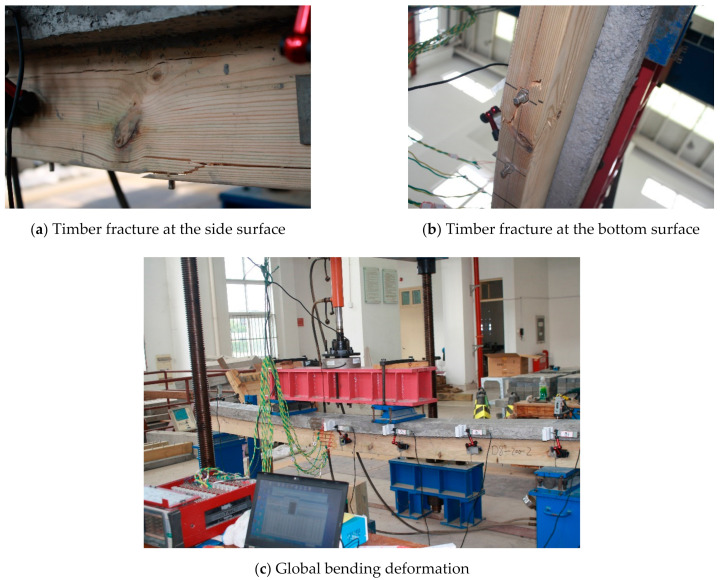
Typical failure modes of the TLC composite beams.

**Figure 8 materials-14-02632-f008:**
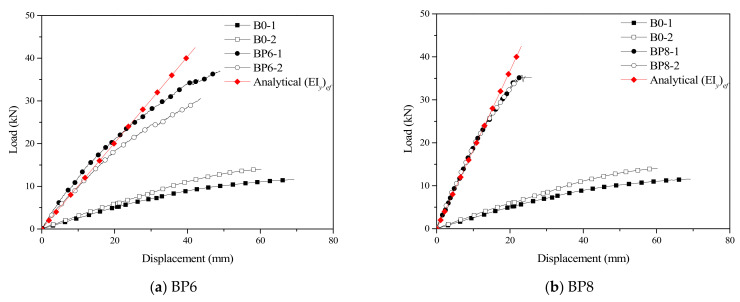
The mid-span displacement curves of the timber beam and the TLC composite beams.

**Figure 9 materials-14-02632-f009:**
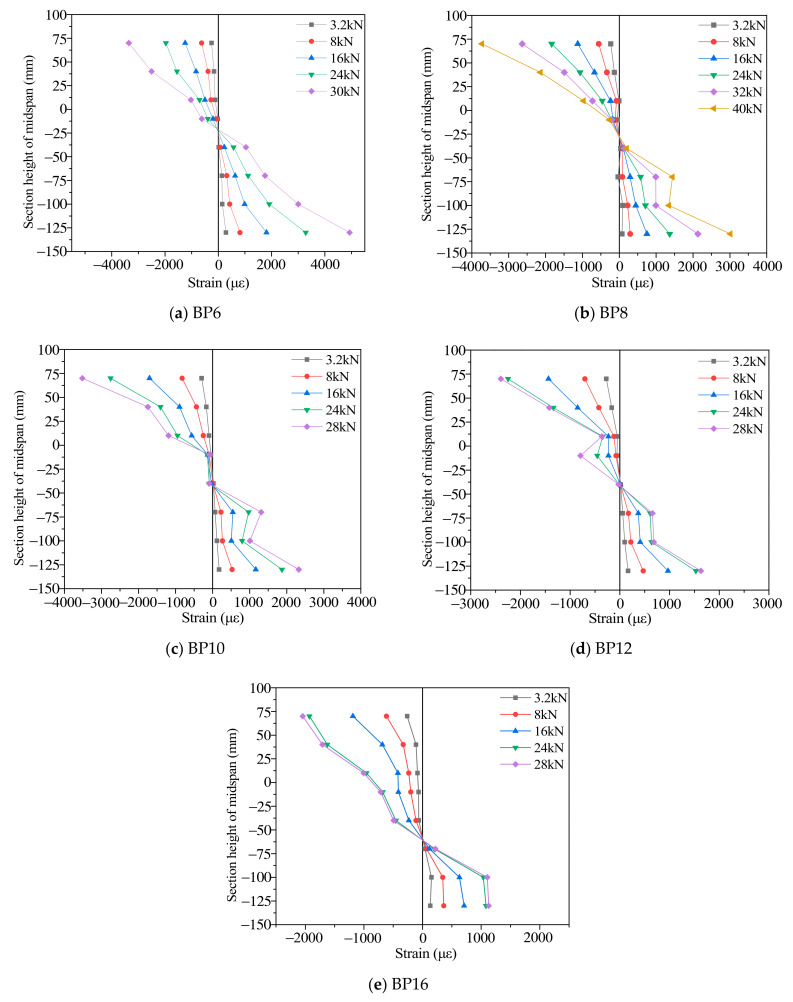
Strain distribution along the height at the mid-span section of the TLC composite beams.

**Figure 10 materials-14-02632-f010:**
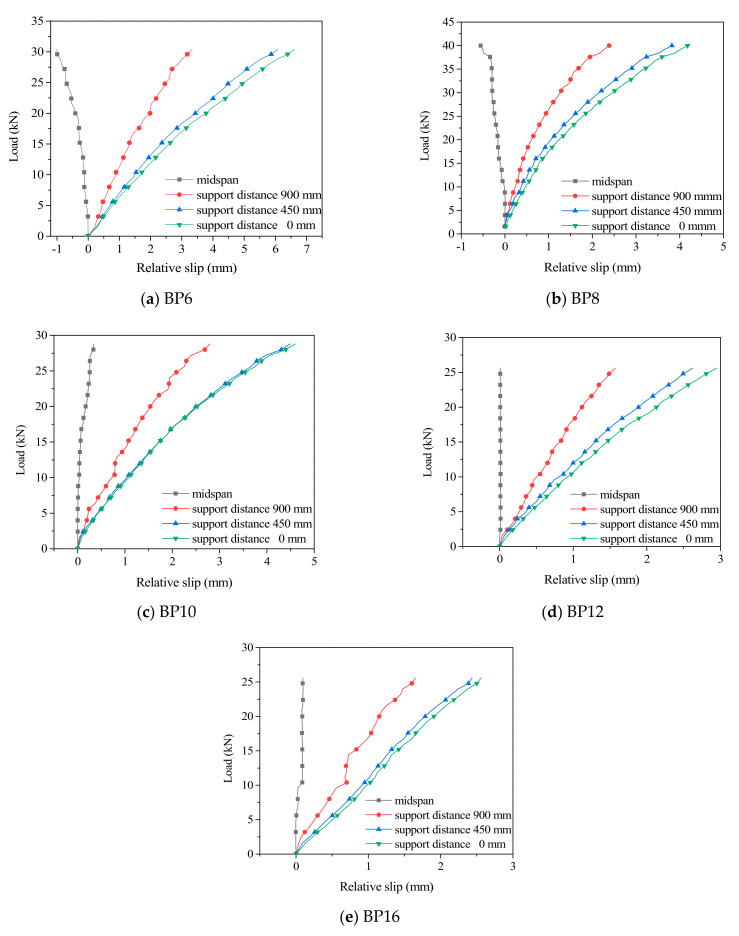
Influence of bolt diameter on the slip of the TLC composite beams.

**Figure 11 materials-14-02632-f011:**
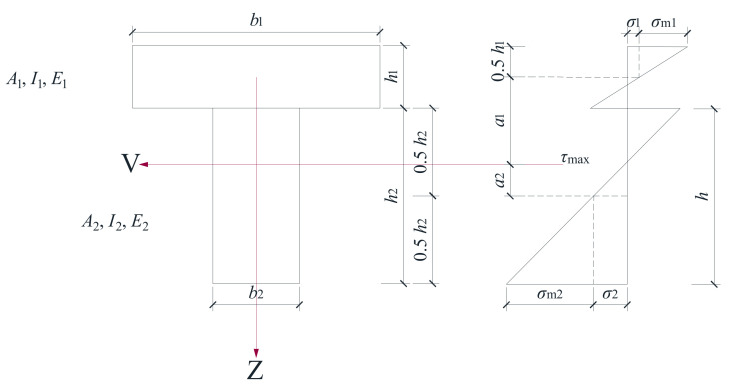
Stress state of the cross−section of the TLC composite beams.

**Table 1 materials-14-02632-t001:** Timber properties.

Timber	Apparent Density (g/cm^3^)	Water Content (%)	Tensile Strength (MPa)	Tensile Strain (με)	Tensile Modulus (GPa)	Compression Strength (MPa)	Compression Strain (με)	Compression Modulus (GPa)
Scotch Pine	0.734	11.2	81.0	8300	10.43	33.4	4480	1.13

**Table 2 materials-14-02632-t002:** Mix proportion of lightweight concrete.

Cement (P·I 42.5) (kg/m^3^)	Fly Ash (kg/m^3^)	Water (kg/m^3^)	Lightweight Aggregate (kg/m^3^)	Water Reducer (kg/m^3^)	Glass Fiber (kg/m^3^)
400	80	260	660	6.0	4.0

**Table 3 materials-14-02632-t003:** Parameters of the TLC composite beams.

Group	Specimen	Concrete		Timber		Section Shape	Bolt Faster		Number of Specimens
		Width (mm)	Height (mm)	Width (mm)	Height (mm)		Diameter (mm)	Spacing (mm)	
B0	B0−1/2/3	/	/	70	140	/	/	/	3
BP6	BP6−1/2/3	200	70	70	140	T shape	6	150	3
BP8	BP8−1/2/3	200	70	70	140	T shape	8	150	3
BP10	BP10−1/2/3	200	70	70	140	T shape	10	150	3
BP12	BP12−1/2/3	200	70	70	140	T shape	12	150	3
BP16	BP16−1/2/3	200	70	70	140	T shape	16	150	3

**Table 4 materials-14-02632-t004:** Test results of the TLC shear specimen with ductile bolts.

Specimen Number	Ultimate Load *P*_u_ (kN)	Ultimate Slip *S*_0_ (mm)	*K_s,_*_0.4_ (kN/mm)	*K_s,_*_0.6_ (kN/mm)	*K_s,_*_0.8_ (kN/mm)	*S_u_* (mm)	*S_y_* (mm)	*D*
SC6−1	27.54	14.42	3.75	1.99	2.12	18.75	11.83	1.58
SC6−2	29.15	15.74	4.12	2.66	1.99	19.72	11.06	1.92
SC6−3	31.27	15.68	3.88	2.84	2.08	19.55	11.12	1.64
Average	29.33	15.27	3.92	2.49	2.07	19.36	11.33	1.71
SC8−1	35.58	12.62	18.97	7.49	3.29	20.95	7.86	2.83
SC8−2	32.11	15.45	22.76	6.95	3.95	22.96	6.94	3.39
SC8−3	37.58	15.86	20.52	7.14	4.36	19.39	7.03	2.44
Average	35.03	14.66	20.76	7.18	3.87	21.05	7.27	2.89
SC10−1	38.39	9.58	13.97	7.01	5.27	16.25	5.22	2.98
SC10−2	41.41	7.77	12.15	6.98	4.92	15.66	6.41	2.29
SC10−3	37.63	8.46	10.22	8.36	5.88	14.23	5.92	2.62
Average	39.17	8.58	12.14	7.48	5.36	15.39	5.84	2.64
SC12−1	56.32	9.51	16.66	11.39	8.66	12.34	5.46	2.68
SC12−2	53.25	9.11	15.09	12.95	8.83	12.83	4.76	2.27
SC12−3	52.22	10.35	15.77	11.39	7.94	11.95	5.03	2.39
Average	53.93	9.64	15.84	11.89	8.48	12.34	5.08	2.43
SC16−1	62.04	6.26	22.97	18.51	14.19	7.64	3.26	2.42
SC16−2	60.03	6.72	23.61	19.25	14.31	8.39	3.24	2.49
SC16−3	58.62	5.92	21.01	18.33	13.82	7.52	3.77	1.98
Average	60.23	6.28	22.52	18.69	14.11	7.85	3.42	2.29

**Table 5 materials-14-02632-t005:** Test results of the control beams and TLC composite beams.

Specimen	*P* = 10 kN			Δ = *L/*300			Timber Cracking State			Ultimate State			*P_L/_*_300_/*P*_max_	*P_su_*/*P*_max_
	Δ (mm)	Stiffness Increase Times	*E*	*P_L/_*_300_ (kN)	Increase Times of *P_L/_*_300_	*E*	*P_su_* (kN)	Δ_su_ (mm)	*E*	*P*_max_ (kN)	Increase Times of P_max_	Δ_max_ (mm)		
B0−1	46.2	—	—	2.14	—	—	—	—	—	11.6	—	58.7	0.18	—
B0−2	37.5	—	—	2.62	—	—	—	—	—	14.0	—	59.2	0.19	—
BP6−1	16.6	2.66	0.53	6.92	3.23	0.62	25.1	44.1	0.43	31.0	2.67	59.5	0.22	0.81
BP6−2	16.2	2.51	0.54	6.84	2.61	0.59	23.8	40.3	0.48	33.2	2.37	62.2	0.21	0.72
BP8−1	14.7	3.26	0.66	7.13	3.33	0.72	25.6	41.6	0.55	26.1	2.25	50.3	0.27	0.98
BP8−2	14.8	3.82	0.62	7.22	2.76	0.69	27.2	40.5	0.57	28.4	2.03	51.8	0.25	0.96
BP10−1	14.2	4.18	0.64	7.76	3.63	0.68	20.5	26.7	0.49	29.6	2.55	44.2	0.26	0.69
BP10−2	13.9	4.62	0.65	8.05	3.07	0.71	18.7	25.4	0.56	30.4	2.17	42.2	0.26	0.62
BP12−1	11.7	4.27	0.56	8.75	4.09	0.63	24.0	33.5	0.41	26.2	2.26	37.6	0.33	0.92
BP12−2	12.6	4.18	0.62	9.22	3.52	0.67	26.2	31.9	0.44	28.4	2.03	35.1	0.32	0.92
BP16−1	7.2	7.38	0.76	13.1	6.12	0.66	37.6	42.4	0.57	40.6	3.50	35.8	0.32	0.93
BP16−2	7.8	6.55	0.71	12.67	4.84	0.65	34.2	40.1	0.53	37.6	2.69	37.2	0.34	0.91

Note: *P* is the load, kN; Δ is the mid-span displacement, mm; the test results of timber beams are used as a reference to evaluate the increase times of the composite beams.

**Table 6 materials-14-02632-t006:** Maximum slip under different load levels.

Specimens	Slip_5kN_ (mm)	*P_L_*_/300_ (kN)	Slip*_L_*_/300_ (mm)	*P_L_*_/250_ (kN)	Slip*_L_*_/250_ (mm)	Ultimate Load (kN)	Slip_UL_ (mm)
BP6−1	0.89	6.92	1.16	7.42	1.29	31.0	6.95
BP6−2	0.87	6.84	1.22	7.05	1.11	33.2	6.85
BP8−1	0.58	7.13	0.78	7.91	0.83	26.1	2.69
BP8−2	0.61	7.22	0.83	7.44	0.82	28.4	2.54
BP10−1	0.58	7.76	0.68	8.63	0.76	29.6	4.74
BP10−2	0.56	8.05	0.74	8.81	0.82	30.4	5.13
BP12−1	0.39	8.75	0.72	9.66	0.81	26.2	2.97
BP12−2	0.42	9.22	0.77	9.34	0.85	28.4	3.15
BP16−1	0.16	13.10	0.29	14.78	0.37	40.6	4.37
BP16−2	0.21	12.67	0.33	14.27	0.41	37.6	4.42

**Table 7 materials-14-02632-t007:** Numerical analysis of the TLC composite beams.

Specimens	Parameters	Serviceability Limit State	Ultimate Limit State	Critical Failure	Unit	Formula
BP6	*k*	15.24	12.75	11.7	kN/mm	(6)
	*γ_b_*	0.11	0.07	0.06	/	(5)
	*a* _1_	95.6	101.12	106.4	mm	(8)
	*a* _2_	11.9	6.38	1.1	mm	(7)
	(*EI*_y_)*_ef_*	2.09 × 10^11^	1.99 × 10^11^	1.96 × 10^11^	kN·mm^2^	(4)
BP8	*k*	5.27	4.42	3.54	kN/mm	(6)
	*γ_b_*	0.21	0.18	0.15	/	(5)
	*a* _1_	90.9	92.67	95.1	mm	(8)
	*a* _2_	16.6	14.83	12.4	mm	(7)
	(*EI*_y_)*_ef_*	3.82 × 10^11^	3.63 × 10^11^	3.29 × 10^11^	kN·mm^2^	(4)
BP10	*k*	4.98	4.24	4.15	kN/mm	(6)
	*γ_b_*	0.24	0.19	0.17	/	(5)
	*a* _1_	89.6	92.24	93.5	mm	(8)
	*a* _2_	17.9	15.26	14.0	mm	(7)
	(*EI*_y_)*_ef_*	3.10 × 10^11^	3.71 × 10^11^	3.76 × 10^11^	kN·mm^2^	(4)
BP12	*k*	3.12	2.67	2.06	kN/mm	(6)
	*γ_b_*	0.34	0.27	0.24	/	(5)
	*a* _1_	87.8	86.94	90.4	mm	(8)
	*a* _2_	19.7	20.56	17.1	mm	(7)
	(*EI*_y_)*_ef_*	4.82 × 10^11^	4.74 × 10^11^	4.67 × 10^11^	kN·mm^2^	(4)
BP16	*k*	2.07	1.70	1.44	kN/mm	
	*γ_b_*	0.40	0.37	0.33	/	(5)
	*a* _1_	78.2	81.34	82.8	mm	(8)
	*a* _2_	29.3	26.16	24.7	mm	(7)
	(*EI*_y_)*_ef_*	5.96 × 10^11^	5.82 × 10^11^	5.72 × 10^11^	kN·mm^2^	(4)

## Data Availability

The data presented in this study are available on request from the corresponding author.

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
