# Peer review of "Experimental Study on Timber−Lightweight Concrete Composite Beams with Ductile Bolt Connectors"

_materials, 2021, doi:10.3390/ma14102632_

Round 1

Reviewer 1 Report

The paper presented an experimental study on a timber-lightweight concrete (TLC) composite beam connected with a ductile connector. The experimental study is well designed and show an interesting results presented in reasonable way. There are small issues to be improves:

  • The quality of the presented figures should be improved e.g. Fig 1 (b, c).
  • There is a problem in the association of most of the references e.g. Page 2 - line 62.
  • The paper needs a through-out proof reading.
  • Further description of the bond properties between concrete and timber would be helpful to include.

Author Response

Responses to Reviewer 1 comments

General comment:

The paper presented an experimental study on a timber-lightweight concrete (TLC) composite beam connected with a ductile connector. The experimental study is well designed and show an interesting results presented in reasonable way. There are small issues to be improves:

Response to general comments:

Thanks for the positive comments and support of the reviewer. We have revised the manuscript carefully as recommended.

Comment 1:

The quality of the presented figures should be improved e.g. Fig 1 (b, c).

Response to comment 1:

Figure 1 (b,c) have been replaced with a higher quality image.

Comment 2:

There is a problem in the association of most of the references e.g. Page 2 - line 62.

Response to comment 2:

All the problems in the association have been modified.

Comment 3:

The paper needs a through-out proof reading.

Response to comment 3:

The paper has been thoroughly revised and checked, and it has been submitted to a professional polishing website for polishing, and the editing certificate is attached at the end of the paper.

Comment 4:

Further description of the bond properties between concrete and timber would be helpful to include.

Response to comment 4:

It can be seen from the push test of shear connectors that the slip between timber and concrete is small, indicating that the timber and concrete with ductile bolt connectors have better bond properties. Relevant expressions have been added in Sections 3.1.1 and 3.1.2.

Reviewer 2 Report

Dear authors,

thank you very much for this article on the necessary and nice topic of timber-lightweight concrete composite beams.

Although at first glance it may seem that your topic is little researched, you will certainly find more very current references around the world. For this reason, I recommend expanding the sources.

I must praise the scope and treatment of the topic, but I found many inconsistencies after reading (see below).

The hypotheses and conclusions are brief and clear. All the results are well described, but I lack more extensive discussion - especially about Table 7.

The scale of your work is a disadvantage in that there is a lot of information, but the reader will soon get lost.

Furthermore, you definitely did not check your text, because there are many errors:
- more specifically, the "Error! Reference source not found." these problem several times,
- typos in the language,
- missing part of the text at the end: "Conflicts of Interest" and "Author Contributions".
- the format of table 1 looks very unprofessional, similar to all other tables - you can stretch the tables to the full width of the page, which would help with the problem of word division - unfortunately, you can see that the article needs a lot of work.
- In the same way, dividing the tables into several pages is not good. 

The article needs to be corrected and improved.

Regards,

Author Response

Responses to Reviewer 2 comments

General comment:

Thank you very much for this article on the necessary and nice topic of timber-lightweight concrete composite beams.

Although at first glance it may seem that your topic is little researched, you will certainly find more very current references around the world. For this reason, I recommend expanding the sources.

I must praise the scope and treatment of the topic, but I found many inconsistencies after reading (see below).

Furthermore, you definitely did not check your text, because there are many errors:

Response to general comments:

Thanks for the comments and support of the reviewer. We have revised the manuscript carefully as recommended. And the references have also been expanded.

Comment 1:

The hypotheses and conclusions are brief and clear. All the results are well described, but I lack more extensive discussion - especially about Table 7.

Response to comment 1:

Thank you for your good advice. We have deepened our discussion of the results, particularly those covered in Table 7 (Page 15, Paragraph 3).

Comment 2:

The scale of your work is a disadvantage in that there is a lot of information, but the reader will soon get lost.

Response to comment 2:

Thank you for your good comments. We also note that this paper is really quite large and covers quite a lot of problems, because it deals with the push-out test of the shear specimens and the bending test of the TLC composite beams, we have streamlined the content of this article, trying to describe the phenomenon and conclusions more clearly, the details are shown in the revised manuscript.

Comment 3:

more specifically, the "Error! Reference source not found." these problem several times.

Response to comment 3:

All the problems in the "Error! Reference source not found" have been modified.

Comment 4:

typos in the language.

Response to comment 4:

We have corrected the typos of the full text, and it has been submitted to a professional polishing website for polishing, and the editing certificate is attached at the end of the paper.

Comment 5:

missing part of the text at the end: "Conflicts of Interest" and "Author Contributions".

Response to comment 5:

The parts of "Conflicts of Interest" and "Author Contributions" have been added at the end of the text.

Comment 6:

the format of table 1 looks very unprofessional, similar to all other tables - you can stretch the tables to the full width of the page, which would help with the problem of word division - unfortunately, you can see that the article needs a lot of work. In the same way, dividing the tables into several pages is not good.

Response to comment 6:

The format of all the tables has been checked and adjusted.

Round 2

Reviewer 2 Report

Dear authors,
Thank you so much for all your corrections and answers.
I would appreciate another round of review, if the editor allows it, where you send the final version without displaying edits.
I would like to read this final version again.

I recommend looking at the Instructions for Authors for better-handled Author Contributions.
I'm not sure that the article is in the journal template.

Best regards,

Author Response

Comment 1:

I would appreciate another round of review, if the editor allows it, where you send the final version without displaying edits.
I would like to read this final version again.

Respond to comment 1:

The final version without displaying edits has been uploaded.

Comment 2:

I recommend looking at the Instructions for Authors for better-handled Author Contributions.
I'm not sure that the article is in the journal template.

Respond to comment 2:

Thank you for your comments. The  Author Contributions have been modified.

Round 3

Reviewer 2 Report

Dear author,
Thank you for your sending the final version. I see that all is fine. The article is recommended for publication.
Regards